# Modeling Renewable Energy Systems by a Self-Evolving Nonlinear Consequent Part Recurrent Type-2 Fuzzy System for Power Prediction

Jafar Tavoosi [1,*](ID), Amir Abolfazl Suratgar [2], Mohammad Bagher Menhaj [2], Amir Mosavi [3,4,*](ID), Ardashir Mohammadzadeh [5](ID) and Ehsan Ranjbar [2]

1   Department of Electrical Engineering, Faculty of Engineering, Ilam University, Ilam, Iran
2   Center of Excellence on Control and Robotics, Department of Electrical Engineering, Amirkabir University of Technology, 424 Hafez Ave, Tehran, Iran; a-suratgar@aut.ac.ir (A.A.S.); menhaj@aut.ac.ir (M.B.M.); en.ranjbar.eeng@gmail.com (E.R.)
3   Faculty of Civil Engineering, Technische Universität Dresden, 01069 Dresden, Germany
4   John von Neumann Faculty of Informatics, Obuda University, 1034 Budapest, Hungary
5   Department of Electrical Engineering, University of Bonab, Bonab 551761167, Iran; a.mzadeh@ubonab.ac.ir
*   Correspondence: j.tavoosi@ilam.ac.ir (J.T.); amir.mosavi@mailbox.tu-dresden.de (A.M.)

**Abstract:** A novel Nonlinear Consequent Part Recurrent Type-2 Fuzzy System (NCPRT2FS) is presented for the modeling of renewable energy systems. Not only does this paper present a new architecture of the type-2 fuzzy system (T2FS) for identification and behavior prognostication of an experimental solar cell set and a wind turbine, but also, it introduces an exquisite technique to acquire an optimal number of membership functions (MFs) and their corresponding rules. Using nonlinear functions in the "Then" part of fuzzy rules, introducing a new mechanism in structure learning, using an adaptive learning rate and performing convergence analysis of the learning algorithm are the innovations of this paper. Another novel innovation is using optimization techniques (including pruning fuzzy rules, initial adjustment of MFs). Next, a solar photovoltaic cell and a wind turbine are deemed as case studies. The experimental data are exploited and the consequent yields emerge as convincing. The root-mean-square-error (RMSE) is less than 0.006 and the number of fuzzy rules is equal to or less than four rules, which indicates the very good performance of the presented fuzzy neural network. Finally, the obtained model is used for the first time for a geographical area to examine the feasibility of renewable energies.

**Keywords:** self-evolving; nonlinear consequent part; convergence analysis; renewable energy; type-2 fuzzy; artificial intelligence; machine learning; big data; data science; fuzzy logic; energy

## 1. Introduction

Renewable energy is expanding rapidly around the world. There are two main reasons for this: one is the issue of fossil fuel pollution and the other is the high cost of fossil fuels. Therefore, research in this field should be developed and supported. One of the powerful tools in data analysis and inference is computational intelligence. Neural networks share lots of significant benefits such as landmark computation ability, parallel processing and adaptation. The fuzzy systems are able to utilize the expert knowledge entitled "if-then rules" and possess actual parameter concepts. As is well known, mathematical modeling is a substantial preliminary step in many control issues. On the other hand, prediction, simulation and modeling of complicated systems established upon physical and chemical principles appear industrious in such a way that they will not yield consolidated mathematical forms [1]. One may suggest system identification as a solution to cope with this problematic issue. This method puts the mathematical equations at the access point, utilizing input-to-output data analysis to increase the efficiency of dynamic process calculations [2]. Computational intelligence lies among the most efficient methods

with excellent fulfillment. Many papers have recently been published on fuzzy modeling and identification. Nonlinear system identification, founded on fuzzy and neuro-fuzzy models, was surveyed [3]. Computational intelligence becomes extremely feasible in the area of renewable energy [4]. For design MPPT control [5], solar water heater selection [6], photovoltaic system failure diagnosis [7] and solar power plant location alternatives [8], computational intelligence has been used. Neural networks were also used by Grahovac et al. [9] in order to model and anticipate bio-ethanol generation from the intermediates and byproducts yielded in the beet-to-sugar procedure. The productivity of the neuro-fuzzy controller in extraction of the maximum yield by flow and energy optimization was demonstrated by Khiareddine et al. [10] in comparison with fuzzy and algorithm controllers. It was asserted that the neuro-fuzzy control system is worthy of being executed in an experimental setup in Tunisia. Ocario et al. [11] testified wind power forecasts in the Portuguese system, exploiting a novel hybrid evolutionary–adaptive methodology. Etemadi et al. [12] predicted the wind power produced by data-driven fuzzy modeling.

Type-2 fuzzy (T2F) logic, which appears more capable and flexible in comparison to type-1, has been investigated for the last ten years. A novel method was suggested for general T2F clustering by Doostparast et al. [13]. Some other applications of T2F sets can be found in textile engineering [14] and aerospace engineering [15]. Fuzzy c-means clustering and high order cognitive map were exerted by Lu in order to model and predict time series by T1FS [16]. T2FS identification has engrossed many researchers [17–23]. Abiyev et al. [17] took advantage of T2F clustering to organize construction of a wavelet TSK-based T2FS. They brought forth an adaptive law to update the parameters of the antecedent part and ultimately, they employed a gradient learning algorithm to bring parameters of the descendant part up to date. T2FSs were applied for elicitation of fuzzy rules and casting derogatory features off [24]. The proposed mechanism took advantage of the self-evolution capability in such a way that identification of the integral structure of the network would become efficient and there would be no requirement for initial start-up of the network structure. The antecedent part and modulation parameters are trained in order to hold parameter learning in the network true, utilizing back-propagation errors. Tuning parameters of the resultant part, the rule-ordered Kalman filter algorithm assists in network sharpness amelioration. The genetic algorithm [25] and PSO [26] are among the learning mechanism of T2F neural networks which have been conversed and scrutinized so far. Research development on T2F systems has brought about their vast usages in various fields such as time-series prediction [27], DC motor control [28], clinical practice guideline encryption [29], pattern recognition [30], robot control [31] and control of nonlinear systems [32,33]. A new smart type of reduction is held forth in [34]. A T2FS is optimized by its type-1 counterpart in [35]. The learning process was held true, merging and extending the type-1 membership functions. Henceforth, the novel constructed T2FS went under implementation on a programmable chip.

It is worth noting that most of the control engineers and system analyzers consider actual systems represented in nonlinear dynamics; not only do these system outputs momentarily turn dependent upon the input, but also, they appear reliant on the delayed inputs/outputs. This leads to a responsible consideration of both external and internal dynamics as a non-negligible essential remark in system modeling. Delayed inputs/outputs have to be used in external dynamics. Another feedback, denoted as "recurrent neuron", has to be exerted in internal dynamics. Wu et al. [36] presented the solution of recurrent FSs for problematic classification. Not only does this paper contribute to minimization of the cost function utilizing a recurrent fuzzy neural network, but it also proposes maximization of the discriminability of adopting a novel approach. Some modern recurrent fuzzy systems are presented in [37]. This special kind of neural network in the resultant part functions input variables in a nonlinear manner. There have hardly been any studies on recurrent T2F systems so far. Some of them are surveyed in the following. A contributive recurrent interval T2FS is presented in order to identify nonlinear systems in [30]. The novel technique requires initial information about plant order and input numbers as well.

Furthermore, the convergence issue in the learning algorithm is not taken into consideration and conversed even theoretically. Juang et al. [15] put forth another contributive recurrent T2F neural network to model dynamical systems. There is not any rule pruning, which leads to extremely overlapped fuzzy sets. Soft switching of the nonlinear model is superior to the linear one in order to identify nonlinear systems [1]. Consequently, our suggested technique is established upon the nonlinear resultant part in fuzzy rules. Rarely may one find comprehensive works on nonlinear consequent parts in fuzzy systems; however, some of the studies in this arena are shortly surveyed in the following. A reduction in the number of rules was carried out by Moodi in a fuzzy system using the TSK fuzzy model accompanied by a nonlinear consequent part [38]. The result of a rule is supposed to comprise a linear term and a nonlinear one. In their attempts, the numerous rules decrease and model precision simultaneously shows an increase at the cost of complication abundance in the fuzzy model. The NFNN was constructed applying fuzzy rules which merge nonlinear functions. The linear consequent part requires more rules to achieve the desired precision during the modeling of complicated nonlinear processes. The increasing number of rules represents the increasing number of neurons [39]. Some recent works on T2F neural networks can be seen in many applications such as 2DOF robot control [40], 3 parallel robots control [41], PMSM control [42], water temperature control [43,44], environmental temperature control [45] and UAV control [46]. Tavoosi and Badamchizadeh [47] proposed a T2S with linear "then part" for dynamic modeling. Their pivotal contribution was rule pruning in such a way that an increase in learning speed would be targeted to attain a reduction in the parameters in both MF parameters and descendant parts. Tavoosi et al. [48,49] have made another contribution to the issue, bringing forth a novel technique for analyzing the stability of one class of T2F systems. Another analysis method for stability was also suggested by Jahangiri et al. [50]. Suratgar and Nikravesh [51] proposed a modern technique of fuzzy linguistic modeling as well as integral stability analysis. In [52], a fuzzy neural network has been used for wind speed forecasting. In [53], a comparison between ANFIS and an autoregressive method for wind speed/power prediction has been performed. In [54], a fuzzy control on the basis of a predictive technique for a governing system has been presented. In [55], a multilayer perceptron is combined with an adaptive fuzzy system to forecast the performance of a wind turbine. Some disadvantages and shortcomings of the works studied above are: lack of convergence proof, long training time (not usable in online applications), high complexity of the model, lack of proper accuracy. On the other hand, so far, no applied research has been conducted to use renewable energies in the Ilam region. Unfortunately, there are no wind turbines in this area, and solar cells have also not been used on a large scale to supply electricity to a neighborhood or even several houses. Due to this issue, the main innovation of this paper is the feasibility study of new energy use in the Ilam region.

Therefore, this paper proposes NCPRT2FS for nonlinear system identification. The nonlinear systems here are the same as solar cells and wind turbines. The objective of identifying the system is to use it to specify the efficiency of the renewable energy system in the Ilam region. The innovations of this article are as follows: (1) Using a nonlinear consequent part in the rules. (2) Introducing a new mechanism in structure learning. (3) Using an adaptive learning rate (different from the other studies in the literature). (4) Convergence analysis of the T2F neural network learning algorithm. Finally, (5) New optimization techniques (including pruning fuzzy rules, initial adjustment of MFs, etc.). The paper is divided into six sections. Section 2 presents a short surveying of T2F logic. Section 3 entails an inspection of the structure of NCPRT2FS. The learning convergence of NCPRT2FS is subsumed relying upon Lyapunov theory in Appendix A. Section 4 presents simulative identification studies, taking into account a solar photovoltaic cell and a wind turbine as the case studies and utilizing their experimental data.

### 2. A Review on T2FSs

Firstly, Zadeh brought forward type-1 fuzzy logic, and introduced the T2F logic in order to provide solutions to some problems of type-1 ten years later. He deemed a fuzzy set where its MF was fuzzy and entitled a "type-2 fuzzy set". T2F sets may typically be exploited when the determination of accurate membership function becomes arduous. For instance, some time series predictions lie among problematic cases, which necessitate the usage of T2F sets. Hence, exploiting T2F sets emerges as advantageous in order to describe some system behaviors.

Certain defects with type-1 fuzzy sets were scrutinized by Castro et al. [56]. Research on T2F systems was limited before the years of 1998. Critical and controversial questions and debate on T2F logic and its usage commenced after publication of a book which contained the solidarity and intersection of T2F sets [57]. Extensive information on T2FS computation, such as defuzzification and type reduction, was suggested by Mendel [58]. A general T2F set, $\widetilde{A}$, may be specified by (1):

$$\widetilde{A} = \int_{x \in X} \mu_{\widetilde{A}}(x)/x = \frac{\int_{x \in X} \left[ \int_{\mu \in J_x} \frac{f_x(\mu)}{\mu} \right]}{x} \tag{1}$$

where $\mu_{\widetilde{A}}(x)$ is a secondary MF; $J_x$ represents the primary membership of $x \in X$, with $\mu \in J_x$; $f_x(\mu) \in [0,1]$ denotes a secondary membership. The primary and secondary MFs in Gaussian form are illustrated in Figure 1.

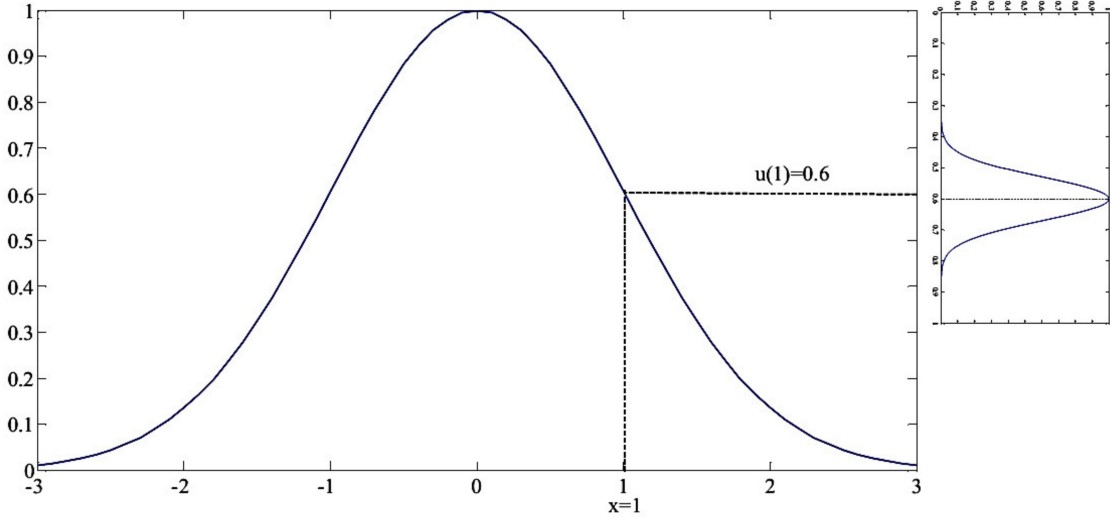

**Figure 1.** Primary and secondary membership functions (MFs).

Note that the secondary MFs lead to interval T2F ones, while $f_x(\mu) = 1$, $\forall \mu \in J_x \subseteq [0,1]$. For more explanation, a crisp number would be fuzzified in two stages supposing that Gaussian MF was exerted to attain a T2F number. First,

$$\mu_1 = \exp\left(-0.5 \cdot \frac{(x-M)^2}{\sigma_x^2}\right) \tag{2}$$

where $\mu_1$ is the primary membership and $M$ and $\sigma_x$ are the primary mean and spread of Gaussian MF, respectively; then,

$$\mu_2(x, \mu_1) = \exp\left(-0.5 \cdot \frac{(a - \mu_1(x))^2}{\sigma_m^2}\right) \tag{3}$$

where $\mu_2(x, \mu_1)$ is the secondary degree, $a \in [0, 1]$ is the domain of the secondary MF for each $x$, and $\sigma_m$ is the secondary spread of the Gaussian MF.

Simple and special kinds of general T2F sets change the same as the interval T2F one. Figure 2 depicts two interval T2F sets. A fuzzy set specified by a Gaussian MF by mean/width $m/[\sigma_1, \sigma_2]$ is demonstrated in Figure 2a. Two T2F sets are given in Figure 2. Figure 2b illustrates a fuzzy set with an MF of Gaussian form encompassing a distinct standard deviation of σ. However, the mean value is quite uncertain and adopts values in the interval of $[m_1, m_2]$.

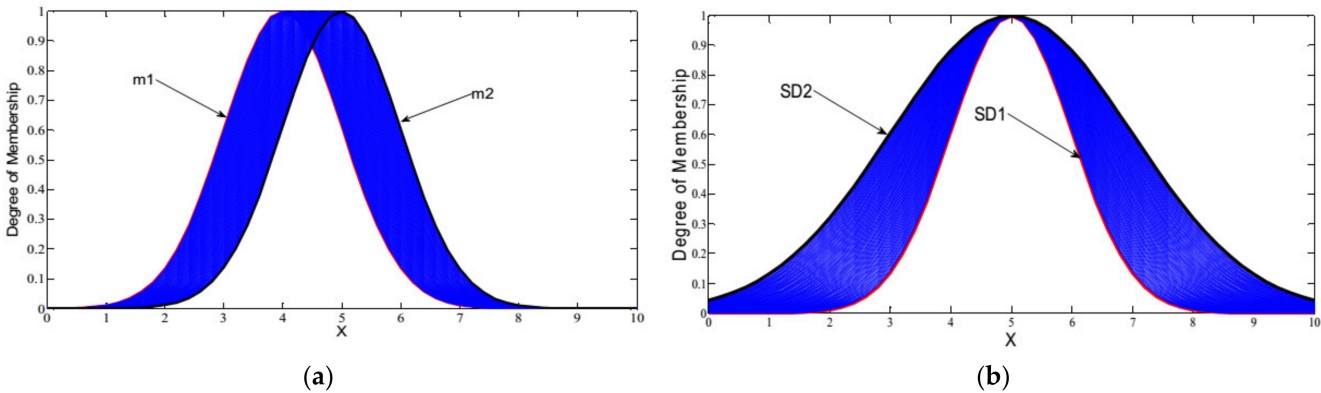

(**a**)  (**b**)

**Figure 2.** (**a**) Uncertainty in width and (**b**) uncertainty in center.

An MF of Gaussian form with determined σ and uncertain *m*, as seen in Figure 2a, is applied through all of this paper.

*Type-2 Fuzzy Systems*

One may gain a certain number by defuzzifying a T1FS [59], whereas T2FS yields a T2F set. This is the reason one has to endeavor to succeed in the reduction in fuzzy set type from two to one in a process entitled "Type Reduction". The process is a challenging issue of high significance in T2F systems [60]. Figure 3 displays the structure of a T2F system.

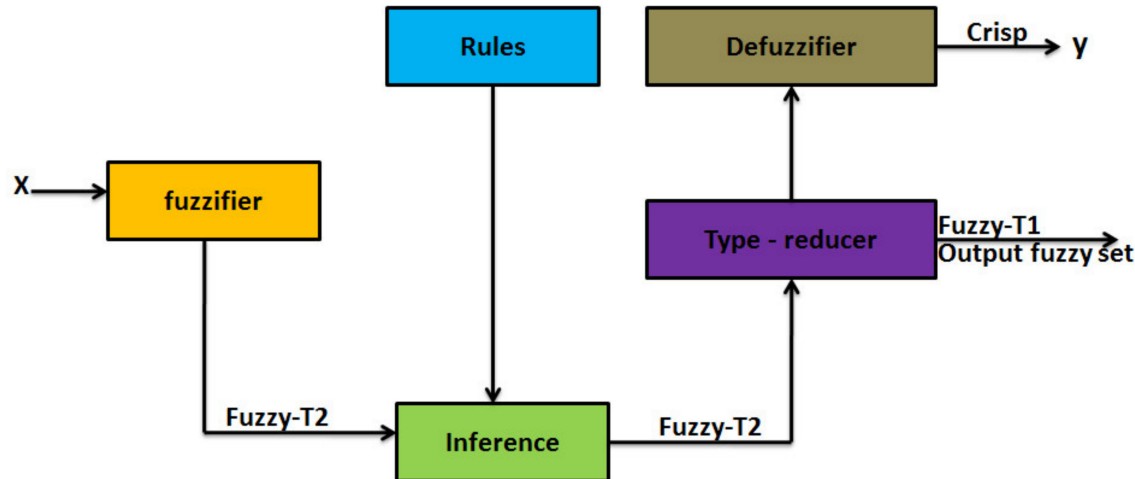

**Figure 3.** The structure of a T2F system.

As can be easily grasped through Figure 4, construction of the T2FS will be the same as the organization of type-1 if the "Type-Reduction" block is neglected.

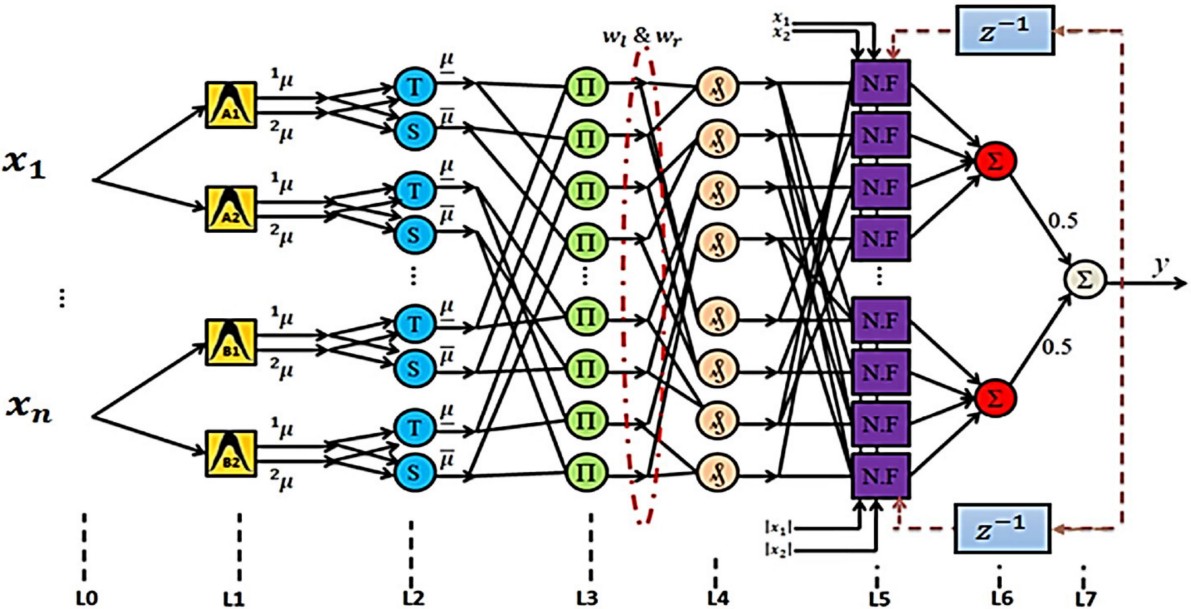

**Figure 4.** The structure of the proposed NCPRT2FS.

## 3. The Proposed NCPRT2FS

Section 3 tries to consolidate the nonlinear descendant or resultant part of recurrent T2F systems into a formula. Taking into account two informative and useful points that are mentioned later, the descriptive equation of (1) establishes the kth rule:

1) TSK-based T2FSs, usually yield a polynomial constructive of the inputs;
2) The outputs are represented by T1F sets [61].

This study recommends a novel NCPRT2FS, of which its total construction is illustrated in Figure 4. As one may see, the system clearly embodies seven layers. Generally speaking, the kth rule would be demonstrated in the following terms in a first-order T2FS with a TSK model by $M$ rules and $n$ inputs:

$$R^k : if \ x_1 \ is \ \widetilde{A}_1^k \ and \dots and \ x_n \ is \ \widetilde{A}_n^k \ then \ \widetilde{y}_k = C_{k,0} + C_{k,1}x_1 + \dots + C_{k,n}x_n$$

where $k = 1, \dots, M$ is the number of rules, $x_i(i = 1, \dots, n)$ are inputs, and $\widetilde{y}_k$ is the output of the $kth$ rule. $\widetilde{y}_k$ is an interval T1F set and $\widetilde{A}_i^k$ are antecedent sets; $C_{k,i} \in [c_{k,i} - s_{k,i}, c_{k,i} + s_{k,i}]$ represent consequent sets, where $c_{k,i}$ represents the center of $C_{k,i}$ and $s_{k,i}$ is the spread of $C_{k,i}$.

In this paper, the nonlinear consequent part is taken into account. The resulting kth rule in NCPRT2FS, which has two antecedent variables and three outputs with delayed time shift ranging from one unit to three in the descendant part, is demonstrated in (2):

$$R^k : if \ x_1 \ is \ \widetilde{A}_1^k \ and \ x_2 \ is \ \widetilde{A}_2^k \ then$$

$$\widetilde{y}_k = C_{k,0} + C_{k,1}x_1 + C_{k,2}x_2 + C_{k,3}y(t-1) + C_{k,4}x_1x_2 + C_{k,5}x_1y(t-1) + C_{k,6}x_2y(t-1)$$

$$+ C_{k,7}x_1^2 + C_{k,8}x_2^2 + C_{k,9}y^2(t-1) + C_{k,10}x_1x_2y(t-1) \tag{4}$$

One may make an extension to fuzzy rule (2) considering $n$ antecedent variables and time-delayed outputs in the descendant part with a delaying shift in time ranging from one unit to m units. $n$ may be designed remarking nonlinearity degree and complexity of the unknown system, which is going to be identified next.

The layers' details are as:

**Layer 0**: This layer denotes the inputs.

**Layer 1**: The outputs of fuzzification are written as:

$$^1\mu_{k,i}\left(x_i, \left[\sigma_{k,i}, {}^1m_{k,i}\right]\right) = e^{-0.5\left(\frac{x_i - {}^1m_{k,i}}{\sigma_{k,i}}\right)^2} \tag{5}$$

$$^2\mu_{k,i}\left(x_i, \left[\sigma_{k,i}, {}^2m_{k,i}\right]\right) = e^{-0.5\left(\frac{x_i - {}^2m_{k,i}}{\sigma_{k,i}}\right)^2} \tag{6}$$

where $m_{k,i} \in \left[{}^1m_{k,i}, {}^2m_{k,i}\right]$ and $\sigma_{k,i}$ are the uncertain mean and spread for *kth* rule and *ith* input.

**Layer 2**: The T-norm and S-norm are computed as:

$$\underline{\mu}_{k,i}(x_i) = {}^1\mu_{k,i}(x_i).^2\mu_{k,i}(x_i), \quad k = 1, 2, \ldots, M, \ i = 1, 2, \ldots, n \tag{7}$$

$$\overline{\mu}_{k,i}(x_i) = {}^1\mu_{k,i}(x_i) + {}^2\mu_{k,i}(x_i) - \underline{\mu}_{k,i}(x_i) \tag{8}$$

**Layer 3**: The rule firings ($\underline{f}^k$ and $\overline{f}^k$) are:

$$\underline{f}^k = \prod_{i=1}^n \underline{\mu}_{k,i} \ ; \ \overline{f}^k = \prod_{i=1}^n \overline{\mu}_{k,i} \tag{9}$$

**Layer 4**: The left-most/right-most firing are obtained as:

$$f_l^k = \frac{\overline{w}_l^k \overline{f}^k + \underline{w}_l^k \underline{f}^k}{\overline{w}_l^k + \underline{w}_l^k} \ ; \ f_r^k = \frac{\overline{w}_r^k \overline{f}^k + \underline{w}_r^k \underline{f}^k}{\overline{w}_r^k + \underline{w}_r^k} \tag{10}$$

where $w$ are adjustable weights.

**Layer 5**: The rule left/right firings are:

$$\begin{aligned} y_l^k = & c_{k,0} + c_{k,1}x_1 + c_{k,2}x_2 + c_{k,3}y(t-1) + c_{k,4}x_1x_2 + c_{k,5}x_1y(t-1) \\ & + c_{k,6}x_2y(t-1) + c_{k,7}x_1^2 + c_{k,8}x_2^2 + c_{k,9}y^2(t-1) \\ & + c_{k,10}x_1x_2y(t-1) - s_{k,0} - s_{k,1}|x_1| - s_{k,2}|x_2| - s_{k,3}|y(t-1)| \\ & - s_{k,4}|x_1x_2| - s_{k,5}|x_1y(t-1)| - s_{k,6}|x_2y(t-1)| - s_{k,7}x_1^2 \\ & - s_{k,8}x_2^2 - s_{k,9}y^2(t-1) - s_{k,10}x_1x_2y(t-1) \end{aligned} \tag{11}$$

$$\begin{aligned} y_r^k = & c_{k,0} + c_{k,1}x_1 + c_{k,2}x_2 + c_{k,3}y(t-1) + c_{k,4}x_1x_2 + c_{k,5}x_1y(t-1) \\ & + c_{k,6}x_2y(t-1) + c_{k,7}x_1^2 + c_{k,8}x_2^2 + c_{k,9}y^2(t-1) \\ & + c_{k,10}x_1x_2y(t-1) + s_{k,0} + s_{k,1}|x_1| + s_{k,2}|x_2| + s_{k,3}|y(t-1)| \\ & + s_{k,4}|x_1x_2| + s_{k,5}|x_1y(t-1)| + s_{k,6}|x_2y(t-1)| + s_{k,7}x_1^2 \\ & + s_{k,8}x_2^2 + s_{k,9}y^2(t-1) + s_{k,10}x_1x_2y(t-1) \end{aligned} \tag{12}$$

**Layer 6**: $\hat{y}_l$ and $\hat{y}_r$ are:

$$\hat{y}_l = \frac{\sum_{k=1}^M f_l^k y_l^k}{\sum_{k=1}^M f_l^k} \tag{13}$$

$$\hat{y}_r = \frac{\sum_{k=1}^M f_r^k y_r^k}{\sum_{k=1}^M f_r^k} \tag{14}$$

**Layer 7**: The output is:

$$\hat{y} = \frac{\hat{y}_l + \hat{y}_r}{2} \tag{15}$$

In this article, structure learning is realized by exploiting T2F clustering. As one knows, an efficacious method is suggested to procreate fuzzy rules in real-time and decrease computations in antecedent part in structure optimization [62]. Structure learning appears

as a great assistance in the simplification of T2FS, taking advantage of the reduction in fuzzy rules. Scrutinizing more, its duty is not only the production of novel membership but also pruning additional MFs and rules. In the input layer, a rule geometrically represents a cluster. Its firing degree could be taken into account as the degree of input data that belongs to a cluster. The center of the firing degree in NCPRT2FS is calculated by (16) since it is an interval.

$$f_k = \frac{\underline{f}^k + \overline{f}^k}{2} \tag{16}$$

Additionally, for generation of a new MF, find:

$$\mu_{\widetilde{A}_i^k} = \frac{\mu_{\widetilde{A}_i^k} + \overline{\mu}_{\widetilde{A}_i^k}}{2} \quad , \quad i = 1, 2, \ldots, n \tag{17}$$

For every incoming data $\overrightarrow{x} = \{x_1, \ldots, x_n\}$, calculate:

$$I = \arg \max_{1 \le k \le M(t)} f_k \tag{18}$$

For newly generated rules:

$$I_i = \arg \max_{1 \le k \le k_i(t)} \mu_{\widetilde{A}_i^k} \quad , \quad i = 1, 2, \ldots, n \tag{19}$$

where $M(t)$ is the existing number of rules at time $t$. If $I \le \varnothing_{th}$, the system generates a new rule, where $\varnothing_{th} \in (0\ 1)$ is a threshold that is defined [63]. If $I_i > \rho$, where $\rho \in [0\ 1]$ is a previously defined threshold, then use the existing fuzzy set $\widetilde{A}_i^{I_i}$ as the antecedent part of the new rule in input variable $i$. Otherwise, one could produce a novel MF in input variable $i$ and hold the equation, $k_i(t+1) = k_i(t) + 1$, true. The number of MFs is defined by the parameter $\rho$ in each input variable. Fuzzy clustering is a technique to structure a fuzzy model [64]. A new T2F clustering technique, which is a development of Krishnapuram and Keller Possibilistic C-Mean (PCM) [65], is suggested and described by:

$$J_m(x, \widetilde{\mu}, c) = min \left[ \sum_{i=1}^{c} \sum_{j=1}^{N} \widetilde{\mu}_{ij}^m D_{ij} + \sum_{i=1}^{c} \eta_i \sum_{j=1}^{N} (1 - \widetilde{\mu}_{ij})^m \right] \tag{20}$$

$$S.T : \begin{cases} 0 < \sum_{j=1}^{N} \widetilde{\mu}_{ij} < N \\ \widetilde{\mu}_{ij} \in [0, 1] \quad \forall i, j \\ \max \widetilde{\mu}_{ij} > 0 \quad \forall j \end{cases} \tag{21}$$

where $\widetilde{\mu}_{ij}$ is type-2 MF in the $j^{th}$ data for the $i^{th}$ cluster. Moreover, the symbols $D_{ij}$, c, and N are the Euclidean distance of the $j^{th}$ data in the $i^{th}$ cluster center, clusters and input data numbers, respectively. $\eta_i$ is also a positive number. $D_{ij}$ has to be as small as possible as the first term. On the other hand, the memberships in a cluster have to be greater as much as possible. They have to stay in the interval of [0, 1] and their sum is confined to become smaller than the number of input data. Equation (21) appears as the descriptive term. That $\eta_i$ corresponds to ith cluster, and is of the order of $D_{ij}$, is greatly welcomed [65]. The distance to the cluster's center must be as low as possible (first term). It is desirable that $\eta_i$ relate to $i^{th}$ cluster and be of the order of $D_{ij}$ [63].

$$\eta_i = \frac{\sum_{j=1}^{N} \widetilde{\mu}_{ij}^m D_{ij}}{\sum_{j=1}^{N} \widetilde{\mu}_{ij}^m} \quad \forall i = 1, \ldots, c$$

Using (20), the optimal values of the centers of the clusters are achieved. The initial $m_{k,i}$ and $\sigma_{k,i}$ for the $k_i(t+1)$ *th* interval T2F set are:

$$m_{k,i} \in [v_i - 0.1v_i,\ v_i + 0.1v_i]$$

$$\sigma_{k_i(t+1)i} = \beta \left| v_i - \frac{^1m_{I_i,i} + {}^2m_{I_i,i}}{2} \right|$$

where $v_i$ is the optimal value of the cluster's center; $\beta > 0$ denotes the degree of overlap between 2 fuzzy sets. In this study, $\beta$ is considered to be 0.5 [61]. The parameters of the consequent part are initialized as:

$$[c_{k,0} - s_{k,0}, c_{k,0} + s_{k,0}] = [yd - 0.1,\ yd + 0.1] \quad , \quad k = 1, 2, \ldots, M \tag{22}$$

where $yd$ is the target signal for input $\vec{x} = \{x_1, \ldots, x_n\}$. All the other consequent parameters are zero.

By repeating the above process for each training dataset, new rules are created one after the other until NCPRT2FS is finally complete. The network output is calculated for each input applied. The calculated output is then compared to the target to obtain an error. Assume that the input–output data pair $\{(x_p : t_p)\}\ \forall p = 1, \ldots, q$, where $p$ represents the data numbers and $x/t$ is the input/output, respectively. The NCPRT2FS output error can be expressed as follows:

$$e_p = t_p - \hat{y}_p, \tag{23}$$

$$E_p = \frac{1}{2}e_p^2 = \frac{1}{2}(t_p - \hat{y}_p)^2 \tag{24}$$

$$E = \sum_{p=1}^{q} E_p \tag{25}$$

The gradient-based learning algorithm is used for updating the parameters. The mathematical relation of the gradient-based update algorithm is as follows.

$$W_{new} = W_{old} - \eta \frac{\partial E}{\partial W}$$

See Appendix A for more details on the parameter update formulation. We choose the initial $\eta$ as:

$$\eta = \frac{1}{max\left| \frac{\partial \hat{y}(k)}{\partial W} \right|^2}$$

After all the data have been applied, the variable learning rate is determined by the following form.

$$\begin{cases} \textbf{if} & \frac{RMSE\ (l)}{RMSE\ (l-1)} < 1 & \rightarrow & \eta(l) = \eta(l-1) \\ \textbf{if} & \frac{RMSE\ (l)}{RMSE\ (l-1)} \geq 1 & \rightarrow & \eta(l) = 0.9 \times \eta(l-1) \end{cases}$$

where $l$ is the number of iterations. The *RMSE* formula is as follows:

$$RMSE = \sqrt{\frac{1}{N} \sum_{p=1}^{N} (t_p - \hat{y}_p)^2}$$

where $t_p$ and $\hat{y}_p$ are actual and model (NCPRT2FS) outputs at $p$ moment, respectively. The total number of data is denoted by $N$.

### 4. Simulation Results

Two real renewable energy systems are used for identification. The structure is shown in Figure 5.

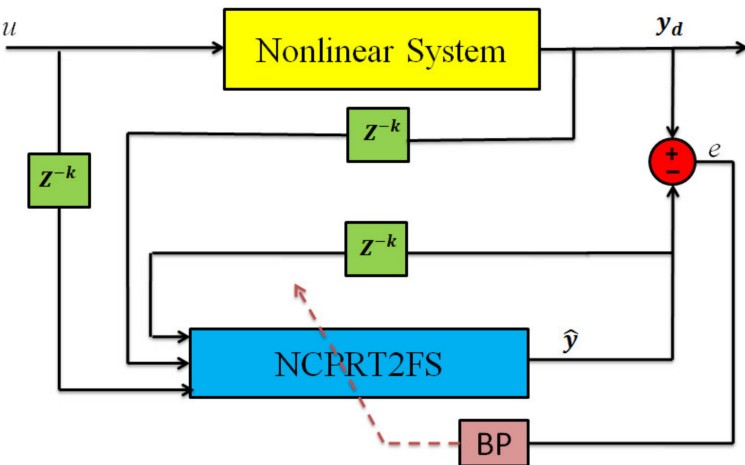

**Figure 5.** The structure of the system and the NCPRT2FS-based identifier.

The inputs to the NCPRT2FS-based identifier are the main input and delayed system output. The parameters of the NCPRT2FS structure should be adjusted to minimize plant output yd and identification yield $\hat{y}$ for all input values of x.

**Example 1:** Real data of a 660kw wind turbine (see Figure 6) have been taken from the Iran Renewable Energy Organization (SUNA) (http://www.suna.org.ir/en/home/ 1 March 2021). The model of the wind turbine is S47-660kw, made by VESTAS (Denmark), and information is given in Table 1.

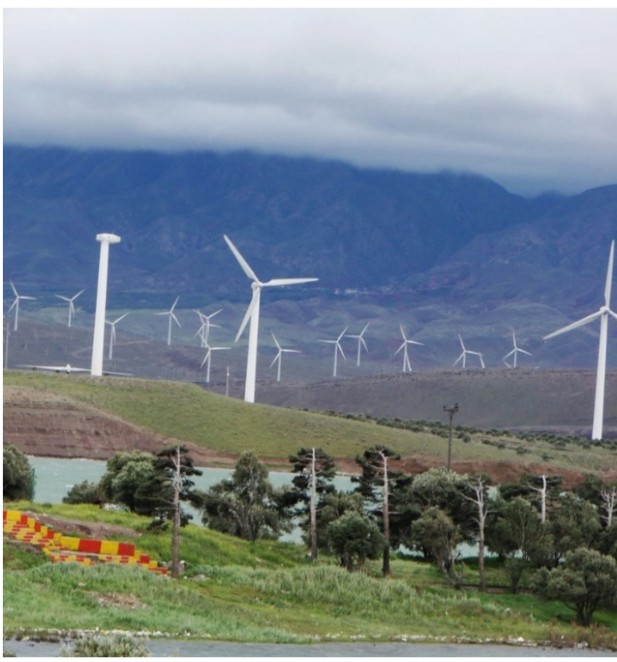

**Figure 6.** Manjil and Rudbar Wind Farm.

In this example, $u(k)$, $k = 1, \ldots, 365$ is wind speed that is fed to the wind turbine system and obtains the 365 samples of $y(k)$, which is the output power of the wind turbine. The other conditions are the same as example 1. Figure 7 exhibits the identification

performance of the NCPRT2FS. Here, the output (solid line) and the NCPRT2FS identifier output (dashed line) are shown.

**Table 1.** Information for Example 1.

| | | | |
|---|---|---|---|
| Cut-in wind speed: | | 4 m/s | |
| Survival wind speed: | | 60 m/s | |
| Rated wind speed: | | 15 m/s | |
| Cut-out wind speed: | | 25 m/s | |
| **Rotor:** | | **Generator:** | |
| Number of blades: | 3 | Type: | Asynchronous |
| Swept area: | 1.735 m$^2$ | Number: | 1.0 |
| Type: | 22.90 | Grid connection: | Thyristor |
| Rotor speed, max: | 28.50 U/min | | |
| Tipspeed: | 70.10 m/s | Voltage: | 400 V |
| Diameter: | 47 m | Speed, max: | 1.650 U/min |
| Material: | GFK | Grid frequency: | 50 Hz |

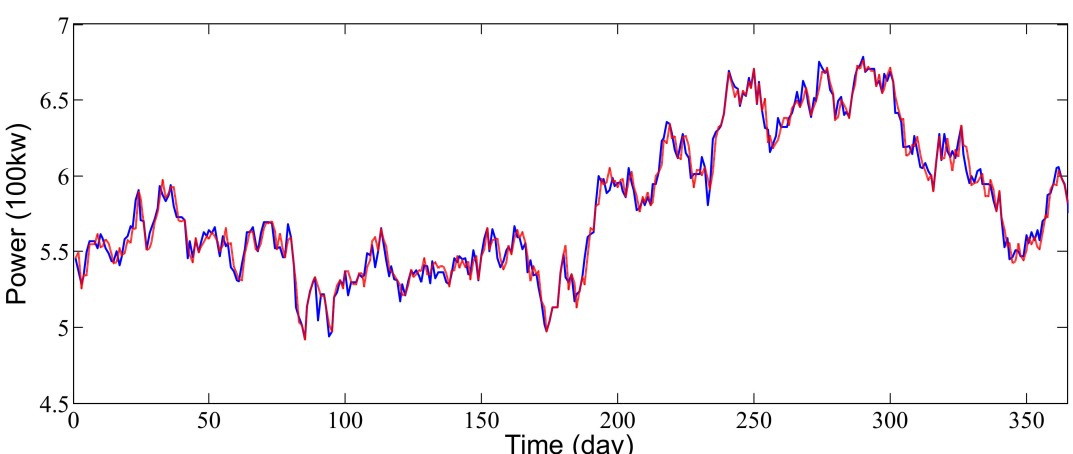

**Figure 7.** Identification performance of the NCPRT2FS for wind turbine.

The trained NCPRT2FS is used to calculate wind power in a place called Ilam (A city in the west of the Islamic Republic of Iran). Figure 8 shows the wind speed of Ilam for a year. Figure 9 shows the predicted wind power in Ilam.

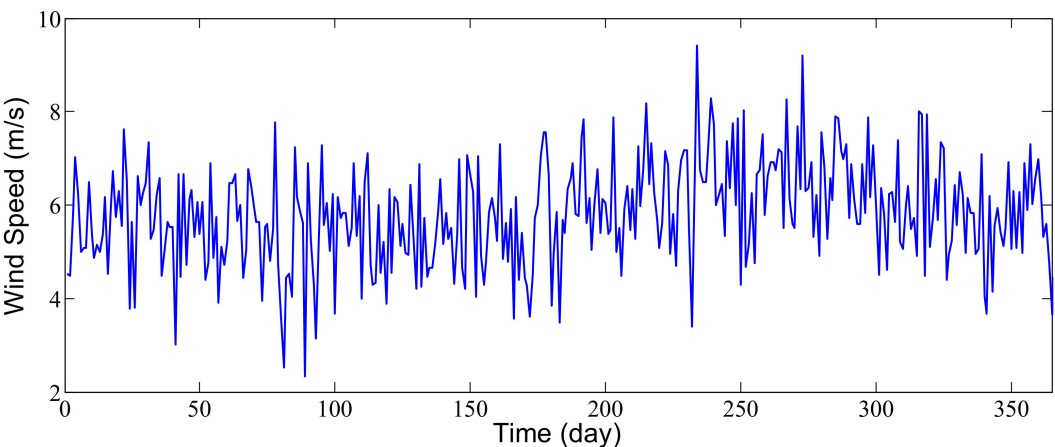

**Figure 8.** Wind speed of a place in Ilam for a year.

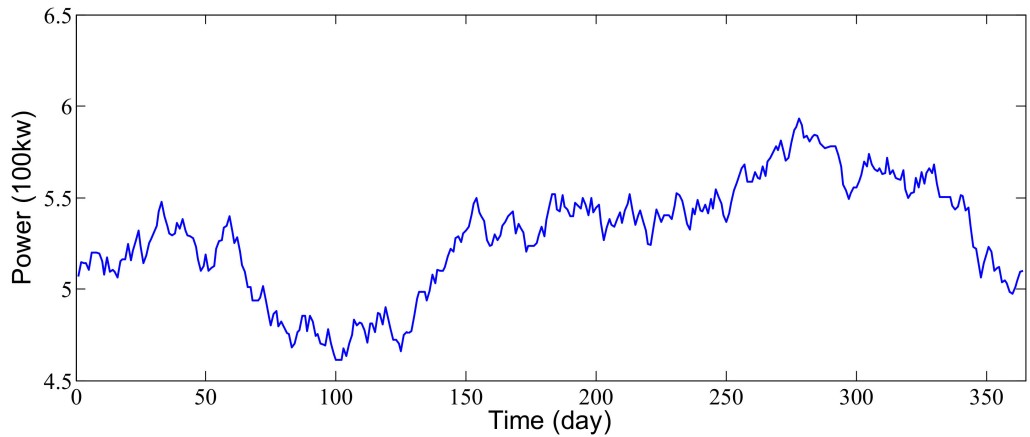

**Figure 9.** Predicted wind power of a place in Ilam for a year.

The final values of the parameters of NCPRT2FS are shown in Table 2.

**Table 2.** The final values of NCPRT2FS parameters.

|  |  | $^1m_{ij}$ | $^2m_{ij}$ | $\sigma_{ij}$ |
|---|---|---|---|---|
| Antecedent parameters | **u(k)** | $^1m_{11} = 3.62$ $^1m_{21} = 6.13$ $^1m_{31} = 8.19$ | $^2m_{11} = 4.32$ $^2m_{21} = 7.02$ $^2m_{31} = 9.51$ | $\sigma_{11} = 0.38$ $\sigma_{21} = 1.10$ $\sigma_{31} = 0.89$ |
|  | **y(k-1)** | $^1m_{12} = 4.93$ $^1m_{22} = 5.34$ $^1m_{32} = 5.81$ $^1m_{42} = 6.11$ | $^2m_{12} = 5.12$ $^2m_{22} = 5.66$ $^2m_{32} = 5.98$ $^2m_{42} = 6.48$ | $\sigma_{12} = 0.21$ $\sigma_{22} = 0.09$ $\sigma_{32} = 0.36$ $\sigma_{42} = 0.18$ |

| fourth layer adaptive weights | | | | |
|---|---|---|---|---|
| $\overline{w}_r^1 = 1.92$ | $\underline{w}_r^1 = 1.50$ | $\overline{w}_l^1 = 1.00$ | $\underline{w}_l^1 = 0.63$ |
| $\overline{w}_r^2 = 1.66$ | $\underline{w}_r^2 = 0.92$ | $\overline{w}_l^2 = 0.71$ | $\underline{w}_l^2 = 0.06$ |
| $\overline{w}_r^3 = 0.80$ | $\underline{w}_r^3 = 0.70$ | $\overline{w}_l^3 = 0.56$ | $\underline{w}_l^3 = 0.43$ |
| $\overline{w}_r^4 = 1.87$ | $\underline{w}_r^4 = 0.94$ | $\overline{w}_l^4 = 0.85$ | $\underline{w}_l^4 = 0.77$ |

| | Rule 1 | Rule 2 | Rule 3 | Rule 4 | Rule 1 | Rule 2 | Rule 3 | Rule 4 |
|---|---|---|---|---|---|---|---|---|
| consequent parameters | $s_{1,0} = 0.40$ | $s_{2,0} = 0.33$ | $s_{3,0} = 0.27$ | $s_{4,0} = 0.52$ | $c_{1,0} = 1.00$ | $c_{2,0} = 1.40$ | $c_{3,0} = 1.00$ | $c_{4,0} = 1.40$ |
| | $s_{1,1} = 0.55$ | $s_{2,1} = 0.39$ | $s_{3,1} = 0.48$ | $s_{4,1} = 0.43$ | $c_{1,1} = 1.10$ | $c_{2,1} = 1.00$ | $c_{3,1} = 1.00$ | $c_{4,1} = 1.00$ |
| | $s_{1,2} = 1.00$ | $s_{2,2} = 1.00$ | $s_{3,2} = 1.00$ | $s_{4,2} = 1.00$ | $c_{1,2} = 1.00$ | $c_{2,2} = 1.32$ | $c_{3,2} = 0.81$ | $c_{4,2} = 0.93$ |
| | $s_{1,3} = 0.43$ | $s_{2,3} = 0.39$ | $s_{3,3} = 0.65$ | $s_{4,3} = 0.90$ | $c_{1,3} = 1.00$ | $c_{2,3} = 1.00$ | $c_{3,3} = 1.65$ | $c_{4,3} = 1.82$ |
| | $s_{1,4} = 0.62$ | $s_{2,4} = 1.00$ | $s_{3,4} = 1.00$ | $s_{4,4} = 1.00$ | $c_{1,4} = 1.00$ | $c_{2,4} = 1.09$ | $c_{3,4} = 1.00$ | $c_{4,4} = 1.00$ |
| | $s_{1,5} = 0.87$ | $s_{2,5} = 0.10$ | $s_{3,5} = 1.00$ | $s_{4,5} = 1.00$ | $c_{1,5} = 1.10$ | $c_{2,5} = 1.00$ | $c_{3,5} = 1.55$ | $c_{4,5} = 1.90$ |
| | $s_{1,6} = 1.00$ | $s_{2,6} = 1.00$ | $s_{3,6} = 1.00$ | $s_{4,6} = 1.00$ | $c_{1,6} = 1.00$ | $c_{2,6} = 1.00$ | $c_{3,6} = 1.00$ | $c_{4,6} = 1.00$ |
| | $s_{1,7} = 0.69$ | $s_{2,7} = 0.66$ | $s_{3,7} = 0.31$ | $s_{4,7} = 0.06$ | $c_{1,7} = 0.80$ | $c_{2,7} = 0.72$ | $c_{3,7} = 0.67$ | $c_{4,7} = 0.81$ |
| | $s_{1,8} = 0.96$ | $s_{2,8} = 0.11$ | $s_{3,8} = 0.54$ | $s_{4,8} = 0.21$ | $c_{1,8} = 1.10$ | $c_{2,8} = 1.00$ | $c_{3,8} = 0.92$ | $c_{4,8} = 0.59$ |
| | $s_{1,9} = 0.30$ | $s_{2,9} = 0.32$ | $s_{3,9} = 0.36$ | $s_{4,9} = 0.98$ | $c_{1,9} = 0.95$ | $c_{2,9} = 0.77$ | $c_{3,9} = 1.00$ | $c_{4,9} = 1.00$ |
| | $s_{1,10} = 0.35$ | $s_{2,10} = 0.31$ | $s_{3,10} = 0.54$ | $s_{4,10} = 0.50$ | $c_{1,10} = 1.00$ | $c_{2,10} = 0.44$ | $c_{3,10} = 0.64$ | $c_{4,10} = 0.89$ |

**Example 2**: A real solar cell system is shown in Figure 10.

In this example, $u(k)$, $k = 1, \ldots, 600$ is solar radiation that is fed to the real solar cell system and 600 samples of $y(k)$ are obtained. The other conditions are the same as in examples 1 and 2. Figure 11 shows the identification performance of the NCPRT2FS for three solar radiations. Here, the plant output (solid line) and the NCPRT2FS identifier output (dashed line) are shown.

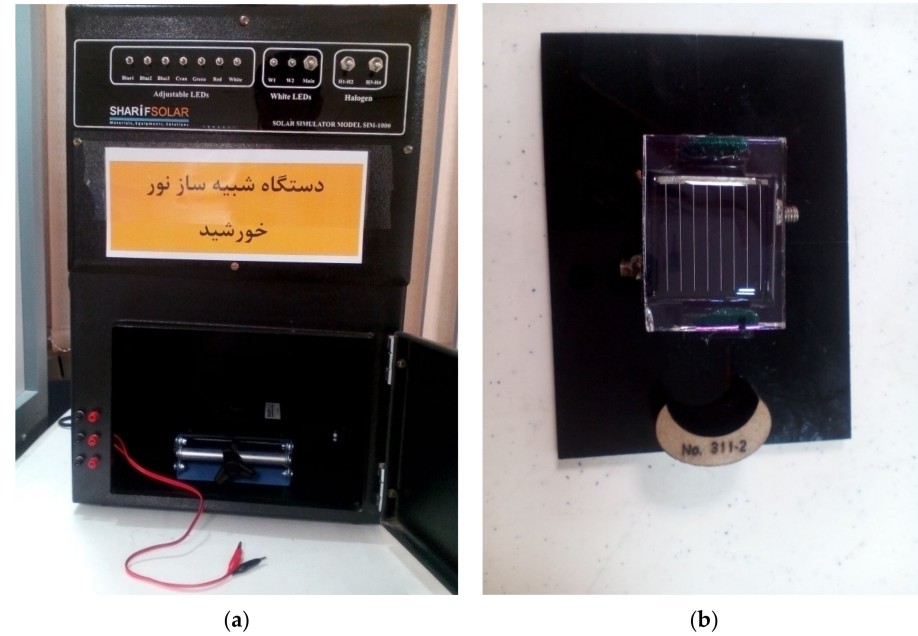

| (**a**) | (**b**) |

**Figure 10.** Experimental solar cell testing system (**a**) and a solar cell (**b**).

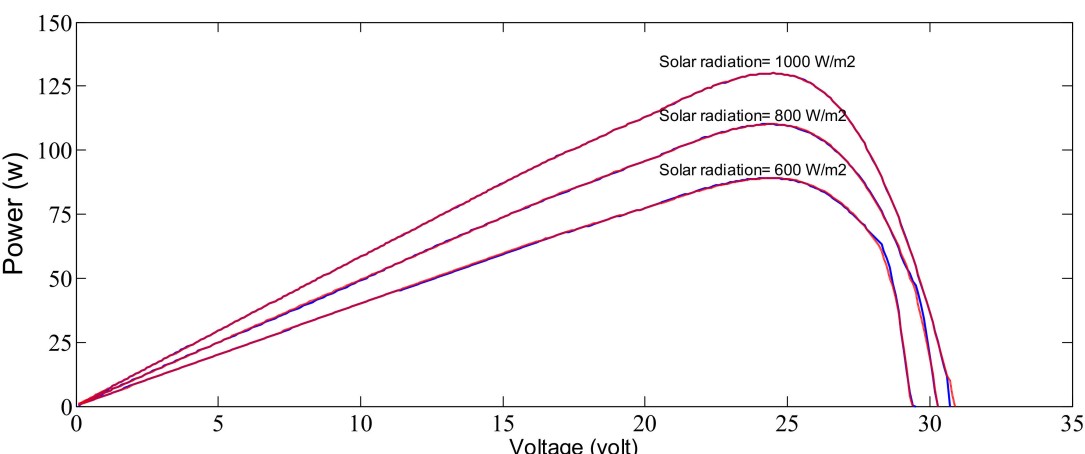

**Figure 11.** Identification results of the NCPRT2FS for solar cell.

After structure learning, for NCPRT2FS, three rules are generated and the RMSE value for the NCPRT2FS and IT2-TSK-FNN for the training and test are shown in Table 3. The final parameters are given in Table 3.

**Table 3.** The final values of NCPRT2FS parameters.

| | | $^1m_{ij}$ | $^2m_{ij}$ | $\sigma_{ij}$ |
|---|---|---|---|---|
| Antecedent parameters | **u(k)** | $^1m_{11} = 251$ $^1m_{21} = 598$ $^1m_{31} = 798$ | $^2m_{11} = 332$ $^2m_{21} = 615$ $^2m_{31} = 949$ | $\sigma_{11} = 43$ $\sigma_{21} = 12$ $\sigma_{31} = 211$ |
| | **y(k-1)** | $^1m_{12} = 69$ $^1m_{22} = 82$ $^1m_{32} = 93$ | $^2m_{12} = 75$ $^2m_{22} = 89$ $^2m_{32} = 97$ | $\sigma_{12} = 11$ $\sigma_{22} = 5$ $\sigma_{32} = 3$ |
| fourth layer adaptive weights | $\overline{w}_r^1 = 0.20$ | $\underline{w}_r^1 = 0.06$ | $\overline{w}_l^1 = 0.12$ | $\underline{w}_l^1 = 0.09$ |
| | $\overline{w}_r^2 = 1.80$ | $\underline{w}_r^2 = 1.00$ | $\overline{w}_l^2 = 1.42$ | $\underline{w}_l^2 = 0.98$ |
| | $\overline{w}_r^3 = 0.57$ | $\underline{w}_r^3 = 0.21$ | $\overline{w}_l^3 = 1.93$ | $\underline{w}_l^3 = 1.10$ |

**Table 3.** *Cont.*

|  | Rule 1 | Rule 2 | Rule 3 | Rule 1 | Rule 2 | Rule 3 |
|---|---|---|---|---|---|---|
| | $s_{1,0} = 0.10$ | $s_{2,0} = 0.84$ | $s_{3,0} = 1.00$ | $c_{1,0} = 0.56$ | $c_{2,0} = 1.00$ | $c_{3,0} = 1.22$ |
| | $s_{1,1} = 0.32$ | $s_{2,1} = 0.39$ | $s_{3,1} = 0.37$ | $c_{1,1} = 0.94$ | $c_{2,1} = 1.60$ | $c_{3,1} = 1.00$ |
| | $s_{1,2} = 1.00$ | $s_{2,2} = 1.00$ | $s_{3,2} = 0.61$ | $c_{1,2} = 1.00$ | $c_{2,2} = 1.00$ | $c_{3,2} = 1.00$ |
| consequent | $s_{1,3} = 0.22$ | $s_{2,3} = 1.20$ | $s_{3,3} = 0.50$ | $c_{1,3} = 1.00$ | $c_{2,3} = 1.77$ | $c_{3,3} = 1.20$ |
| parameters | $s_{1,4} = 0.10$ | $s_{2,4} = 0.42$ | $s_{3,4} = 1.00$ | $c_{1,4} = 1.61$ | $c_{2,4} = 0.60$ | $c_{3,4} = 1.63$ |
| | $s_{1,5} = 0.47$ | $s_{2,5} = 1.00$ | $s_{3,5} = 1.00$ | $c_{1,5} = 1.30$ | $c_{2,5} = 1.00$ | $c_{3,5} = 2.00$ |
| | $s_{1,6} = 0.10$ | $s_{2,6} = 1.00$ | $s_{3,6} = 1.00$ | $c_{1,6} = 1.00$ | $c_{2,6} = 1.11$ | $c_{3,6} = 1.00$ |
| | $s_{1,7} = 1.20$ | $s_{2,7} = 1.00$ | $s_{3,7} = 0.19$ | $c_{1,7} = 1.10$ | $c_{2,7} = 1.50$ | $c_{3,7} = 0.88$ |
| | $s_{1,8} = 1.00$ | $s_{2,8} = 0.36$ | $s_{3,8} = 0.69$ | $c_{1,8} = 1.60$ | $c_{2,8} = 0.89$ | $c_{3,8} = 0.91$ |
| | $s_{1,9} = 1.00$ | $s_{2,9} = 0.28$ | $s_{3,9} = 0.11$ | $c_{1,9} = 1.53$ | $c_{2,9} = 0.95$ | $c_{3,9} = 0.48$ |
| | $s_{1,10} = 0.55$ | $s_{2,10} = 0.35$ | $s_{3,10} = 0.50$ | $c_{1,10} = 0.88$ | $c_{2,10} = 1.00$ | $c_{3,10} = 1.00$ |

The trained NCPRT2FS is used to calculate the solar power of Ilam. Figure 12 shows the solar radiation of Ilam for a year. Figure 13 shows the predicted solar power in Ilam.

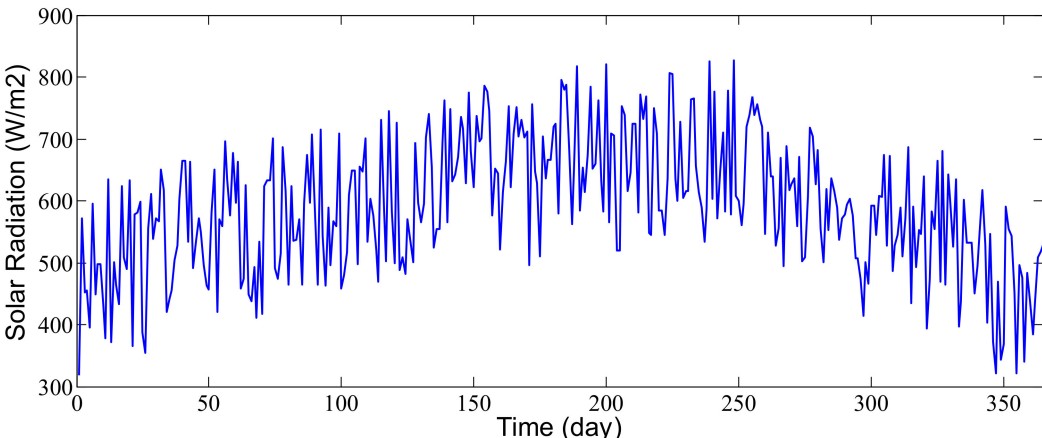

**Figure 12.** Solar radiation of Ilam.

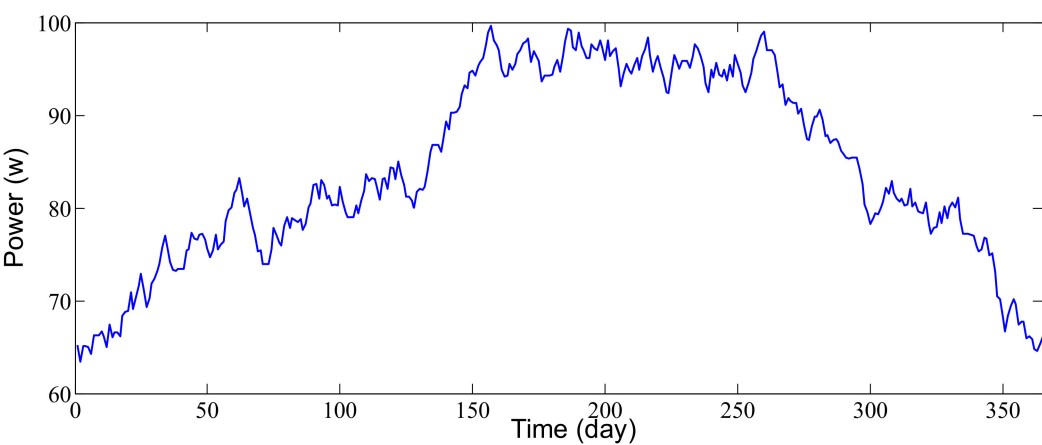

**Figure 13.** Predicted solar power in Ilam for a year.

Table 4 presents the comparison of our proposed method with another method (method of [46]).

**Table 4.** Comparison between results of the proposed method and the method of [46].

| Example | Method of [46] | | | | Proposed NCPRT2FS | | | |
|---|---|---|---|---|---|---|---|---|
| | Rules | Epochs | Run Time (s) | RMSE | Rules | Epochs | Run Time (s) | RMSE |
| 1 | 4 | 34 | 4 | 0.0159 | 4 | 31 | 6 | 0.0057 |
| 2 | 5 | 27 | 4 | 0.00759 | 3 | 39 | 7 | 0.0013 |

Simulations verify that the presented NCPRT2FS has high performances in function approximation and system identification. Table 4 shows that the number of rules of the proposed NCPRT2FS is almost less than the method of [53]; accuracy of identification is better than [53], but the training time in 10 runs (MATLAB 2011a; Dual CPU T3200 @ 2.00; RAM: 2.00 GB; GHz 2.00 GHz) is more than [53]. The references [23,46] presented two different T2F neural structures. They have also been used and evaluated only to identify some theory systems. In the present paper, however, the T2F neural network structure is different from references [23] and [53] and several experimental energy systems have been used for modeling.

## 5. Conclusions

In this paper, a novel Nonlinear Consequent Part Recurrent T2FS (NCPRT2FS) for identification and prediction of renewable energy systems was proposed. The nonlinear consequent part helps to better model highly nonlinear systems. Recurrent structure is a useful choice for the identification of dynamical systems. The self-evolving structure helps to obtain a simpler structure of the NCPRT2FS by ending up with a minimum number of fuzzy sets and fuzzy rules in the end. Simulations showed that the NCPRT2FS based on the backpropagation algorithm and adaptive optimization rate performs better than IT2-TSK-FNN [53] in identification. An S47-660 kw wind turbine (VESTAS company Denmark) and a solar cell were selected as case studies. After data gathering, the proposed method was finally used with the experimental data for the purpose of identification. The RMSE was less than 0.006 and the number of fuzzy rules was equal and less than 4 rules; therefore, the results easily demonstrated the remarkable capability of the NCPRT2FS developed in the paper. In order to continue the work and look to the future, we can use the evolutionary algorithms as a complement to the proposed method for the development of the fuzzy neural network (to increase accuracy, increase convergence, etc.). Different case studies (types of solar cells, types of wind turbines, etc.) should be identified and the appropriate renewable system can be extracted for each geographical location.

**Author Contributions:** Writing—Original draft preparation, J.T., A.A.S., M.B.M., A.M. (Ardashir Mohammadzadeh), E.R., A.M. (Amir Mosavi); Software, J.T., A.A.S., M.B.M., A.M. (Ardashir Mohammadzadeh), E.R., A.M. (Amir Mosavi); Validation, J.T., A.A.S., M.B.M., A.M. (Ardashir Mohammadzadeh), E.R., A.M. (Amir Mosavi); Supervision, J.T., A.M. (Amir Mosavi). All authors have read and agreed to the published version of the manuscript.

**Funding:** This research received Open Access Funding by the Publication Fund of the TU Dresden.

**Institutional Review Board Statement:** Not applicable.

**Informed Consent Statement:** Not applicable.

**Data Availability Statement:** The data are available from via, (j.tavoosi@ilam.ac.ir), (accessed on 1 March 2021), upon reasonable request.

**Conflicts of Interest:** The authors declare that they have no conflict of interest.

## Appendix A

To update the consequent part parameters, Equations (A1)–(A20) are used.

$$^{new}c_{k,0} = {}^{old}c_{k,0} + \eta \cdot 0.5 \cdot e_p \cdot \left[ \frac{f_l^k}{\sum_{k=1}^M f_l^k} + \frac{f_r^k}{\sum_{k=1}^M f_r^k} \right] \tag{A1}$$

$$^{new}c_{k,i} = {}^{old}c_{k,i} + \eta \cdot 0.5 \cdot e_p \cdot \left[ \frac{f_l^k}{\sum_{k=1}^M f_l^k} + \frac{f_r^k}{\sum_{k=1}^M f_r^k} \right] \cdot x_i \quad i = 1, 2 \tag{A2}$$

$$^{new}c_{k,3} = {}^{old}c_{k,3} + \eta \cdot 0.5 \cdot e_p \cdot \left[ \frac{f_l^k}{\sum_{k=1}^M f_l^k} + \frac{f_r^k}{\sum_{k=1}^M f_r^k} \right] \cdot y(t-1) \tag{A3}$$

$$^{new}c_{k,4} = {}^{old}c_{k,4} + \eta \cdot 0.5 \cdot e_p \cdot \left[ \frac{f_l^k}{\sum_{k=1}^M f_l^k} + \frac{f_r^k}{\sum_{k=1}^M f_r^k} \right] \cdot x_1 \cdot x_2 \tag{A4}$$

$$^{new}c_{k,5} = {}^{old}c_{k,5} + \eta \cdot 0.5 \cdot e_p \cdot \left[ \frac{f_l^k}{\sum_{k=1}^M f_l^k} + \frac{f_r^k}{\sum_{k=1}^M f_r^k} \right] \cdot x_1 \cdot y(t-1) \tag{A5}$$

$$^{new}c_{k,6} = {}^{old}c_{k,6} + \eta \cdot 0.5 \cdot e_p \cdot \left[ \frac{f_l^k}{\sum_{k=1}^M f_l^k} + \frac{f_r^k}{\sum_{k=1}^M f_r^k} \right] \cdot x_2 \cdot y(t-1) \tag{A6}$$

$$^{new}c_{k,7} = {}^{old}c_{k,7} + \eta \cdot 0.5 \cdot e_p \cdot \left[ \frac{f_l^k}{\sum_{k=1}^M f_l^k} + \frac{f_r^k}{\sum_{k=1}^M f_r^k} \right] \cdot x_1^2 \tag{A7}$$

$$^{new}c_{k,8} = {}^{old}c_{k,8} + \eta \cdot 0.5 \cdot e_p \cdot \left[ \frac{f_l^k}{\sum_{k=1}^M f_l^k} + \frac{f_r^k}{\sum_{k=1}^M f_r^k} \right] \cdot x_2^2 \tag{A8}$$

$$^{new}c_{k,9} = {}^{old}c_{k,9} + \eta \cdot 0.5 \cdot e_p \cdot \left[ \frac{f_l^k}{\sum_{k=1}^M f_l^k} + \frac{f_r^k}{\sum_{k=1}^M f_r^k} \right] \cdot y^2(t-1) \tag{A9}$$

$$^{new}c_{k,10} = {}^{old}c_{k,10} + \eta \cdot 0.5 \cdot e_p \cdot \left[ \frac{f_l^k}{\sum_{k=1}^M f_l^k} + \frac{f_r^k}{\sum_{k=1}^M f_r^k} \right] \cdot x_1 \cdot x_2 \cdot y(t-1) \tag{A10}$$

$$^{new}s_{k,0} = {}^{old}s_{k,0} + \eta \cdot 0.5 \cdot e_p \cdot \left[ \frac{f_l^k}{\sum_{k=1}^M f_l^k} - \frac{f_r^k}{\sum_{k=1}^M f_r^k} \right] \tag{A11}$$

$$^{new}s_{k,i} = {}^{old}s_{k,i} + \eta \cdot 0.5 \cdot e_p \cdot \left[ \frac{f_l^k}{\sum_{k=1}^M f_l^k} - \frac{f_r^k}{\sum_{k=1}^M f_r^k} \right] \cdot |x_i| \quad i = 1, 2 \tag{A12}$$

$$^{new}s_{k,3} = {}^{old}s_{k,3} + \eta \cdot 0.5 \cdot e_p \cdot \left[ \frac{f_l^k}{\sum_{k=1}^M f_l^k} - \frac{f_r^k}{\sum_{k=1}^M f_r^k} \right] \cdot |y(t-1)| \tag{A13}$$

$$^{new}s_{k,4} = {}^{old}s_{k,4} + \eta \cdot 0.5 \cdot e_p \cdot \left[ \frac{f_l^k}{\sum_{k=1}^M f_l^k} - \frac{f_r^k}{\sum_{k=1}^M f_r^k} \right] \cdot |x_1 x_2| \tag{A14}$$

$$^{new}s_{k,5} = {}^{old}s_{k,5} + \eta \cdot 0.5 \cdot e_p \cdot \left[ \frac{f_l^k}{\sum_{k=1}^M f_l^k} - \frac{f_r^k}{\sum_{k=1}^M f_r^k} \right] \cdot |x_1 \cdot y(t-1)| \tag{A15}$$

$$^{new}s_{k,6} = {}^{old}s_{k,6} + \eta \cdot 0.5 \cdot e_p \cdot \left[ \frac{f_l^k}{\sum_{k=1}^M f_l^k} - \frac{f_r^k}{\sum_{k=1}^M f_r^k} \right] \cdot |x_2 \cdot y(t-1)| \tag{A16}$$

$$^{new}s_{k,7} = {}^{old}s_{k,7} + \eta \cdot 0.5 \cdot e_p \cdot \left[ \frac{f_l^k}{\sum_{k=1}^M f_l^k} - \frac{f_r^k}{\sum_{k=1}^M f_r^k} \right] \cdot x_1^2 \tag{A17}$$

$$^{new}s_{k,8} = {}^{old}s_{k,8} + \eta \cdot 0.5 \cdot e_p \cdot \left[ \frac{f_l^k}{\sum_{k=1}^{M} f_l^k} - \frac{f_r^k}{\sum_{k=1}^{M} f_r^k} \right] \cdot x_2^2 \tag{A18}$$

$$^{new}s_{k,9} = {}^{old}s_{k,9} + \eta \cdot 0.5 \cdot e_p \cdot \left[ \frac{f_l^k}{\sum_{k=1}^{M} f_l^k} - \frac{f_r^k}{\sum_{k=1}^{M} f_r^k} \right] \cdot y^2(t-1) \tag{A19}$$

$$^{new}s_{k,10} = {}^{old}s_{k,10} + \eta \cdot 0.5 \cdot e_p \cdot \left[ \frac{f_l^k}{\sum_{k=1}^{M} f_l^k} - \frac{f_r^k}{\sum_{k=1}^{M} f_r^k} \right] \cdot |x_1 \cdot x_2 \cdot y(t-1)| \tag{A20}$$

The learning rate is indicated by $\eta$.

To update the left and right weights, Equations (A21)–(A24) are used.

$$^{new}\underline{w}_l^k = {}^{old}\underline{w}_l^k + \eta \cdot 0.5 \cdot e_p \cdot \frac{y_l^k - \hat{y}_l}{\sum_{j=1}^{M} f_l^j} \cdot \frac{\underline{f}^k - f_l^k}{\overline{w}_l^k + \underline{w}_l^k} \tag{A21}$$

$$^{new}\overline{w}_l^k = {}^{old}\overline{w}_l^k + \eta \cdot 0.5 \cdot e_p \cdot \frac{y_l^k - \hat{y}_l}{\sum_{j=1}^{M} f_l^j} \cdot \frac{\overline{f}^k - f_l^k}{\overline{w}_l^k + \underline{w}_l^k} \tag{A22}$$

$$^{new}\underline{w}_r^k = {}^{old}\underline{w}_r^k + \eta \cdot 0.5 \cdot e_p \cdot \frac{y_r^k - \hat{y}_r}{\sum_{j=1}^{M} f_r^j} \cdot \frac{\underline{f}^k - f_r^k}{\overline{w}_r^k + \underline{w}_r^k} \tag{A23}$$

$$^{new}\overline{w}_r^k = {}^{old}\overline{w}_r^k + \eta \cdot 0.5 \cdot e_p \cdot \frac{y_r^k - \hat{y}_r}{\sum_{j=1}^{M} f_r^j} \cdot \frac{\overline{f}^k - f_r^k}{\overline{w}_r^k + \underline{w}_r^k} \tag{A24}$$

Finally, the equations for updating the antecedent parameters can be described as follows:

$$^1m_{k,i}^{new} = {}^1m_{k,i}^{old} + \eta \cdot 0.5 \cdot e_p \cdot \left[ \frac{y_l^k - \hat{y}_l}{\sum_{j=1}^{M} f_l^j} \cdot \frac{\partial f_l^k}{\partial^1 m_{k,i}} + \frac{y_r^k - \hat{y}_r}{\sum_{j=1}^{M} f_r^j} \cdot \frac{\partial f_r^k}{\partial^1 m_{k,i}} \right] \tag{A25}$$

$$^2m_{k,i}^{new} = {}^2m_{k,i}^{old} + \eta \cdot 0.5 \cdot e_p \cdot \left[ \frac{y_l^k - \hat{y}_l}{\sum_{j=1}^{M} f_l^j} \cdot \frac{\partial f_l^k}{\partial^2 m_{k,i}} + \frac{y_r^k - \hat{y}_r}{\sum_{j=1}^{M} f_r^j} \cdot \frac{\partial f_r^k}{\partial^2 m_{k,i}} \right] \tag{A26}$$

$$\sigma_{k \cdot i}^{new} = \sigma_{k \cdot i}^{old} + \eta \cdot 0.5 \cdot e_p \cdot \left[ \frac{y_l^k - \hat{y}_l}{\sum_{j=1}^{M} f_l^j} \cdot \frac{\partial f_l^k}{\partial \sigma_{k,i}} + \frac{y_r^k - \hat{y}_r}{\sum_{j=1}^{M} f_r^j} \cdot \frac{\partial f_r^k}{\partial \sigma_{k,i}} \right] \tag{A27}$$

where

$$\frac{\partial f_l^k}{\partial^1 m_{k,i}} = \frac{\overline{w}_l^k \cdot \left[ \overline{f}^k - {}^2\mu_{k,i} \cdot \prod_{l=1, l \neq i}^{n} \left( \overline{\mu}_{k,l} \right) \right] + \underline{w}_l^k \cdot \underline{f}^k}{\overline{w}_l^k + \underline{w}_l^k} \cdot \frac{x_i - {}^1m_{k,i}}{(\sigma_{k,i})^2}, \tag{A28}$$

$$\frac{\partial f_l^k}{\partial^2 m_{k,i}} = \frac{\overline{w}_l^k \cdot \left[ \overline{f}^k - {}^1\mu_{k,i} \cdot \prod_{l=1, l \neq i}^{n} \left( \overline{\mu}_{k,l} \right) \right] + \underline{w}_l^k \cdot \underline{f}^k}{\overline{w}_l^k + \underline{w}_l^k} \cdot \frac{x_i - {}^2m_{k,i}}{(\sigma_{k,i})^2} \tag{A29}$$

$$\frac{\partial f_l^k}{\partial \sigma_{k,i}} = \frac{\overline{w}_l^k \cdot \left[ \left( \overline{f}^k - {}^2\mu_{k,i} \cdot \prod_{l=1, l \neq i}^{n} \left( \overline{\mu}_{k,l} \right) \right) \cdot \frac{(x_i - {}^1m_{k,i})^2}{(\sigma_{k,i})^3} \right]}{\overline{w}_l^k + \underline{w}_l^k}$$

$$+ \frac{\overline{w}_l^k \cdot \left[ \left( \overline{f}^k - {}^1\mu_{k,i} \cdot \prod_{l=1, l \neq i}^{n} \left( \overline{\mu}_{k,l} \right) \right) \cdot \frac{(x_i - {}^2m_{k,i})^2}{(\sigma_{k,i})^3} \right]}{\overline{w}_l^k + \underline{w}_l^k}$$

$$+ \frac{\underline{w}_l^k \cdot \underline{f}^k \cdot \left[ \frac{\left(x_i - {}^1m_{k,i}\right)^2 + \left(x_i - {}^2m_{k,i}\right)^2}{\left(\sigma_{k,i}\right)^3} \right]}{\overline{w}_l^k + \underline{w}_l^k} \tag{A30}$$

$$\frac{\partial f_r^k}{\partial {}^1m_{k,i}} = \frac{\overline{w}_r^k \cdot \left[ \overline{f}^k - {}^2\mu_{k,i} \cdot \prod_{l=1, l \neq i}^n \left( \overline{\mu}_{k,l} \right) \right] + \underline{w}_r^k \cdot \underline{f}^k}{\overline{w}_r^k + \underline{w}_r^k} \cdot \frac{x_i - {}^1m_{k,i}}{\left(\sigma_{k,i}\right)^2} \tag{A31}$$

$$\frac{\partial f_r^k}{\partial {}^2m_{k,i}} = \frac{\overline{w}_r^k \cdot \left[ \overline{f}^k - {}^1\mu_{k,i} \cdot \prod_{l=1, l \neq i}^n \left( \overline{\mu}_{k,l} \right) \right] + \underline{w}_r^k \cdot \underline{f}^k}{\overline{w}_r^k + \underline{w}_r^k} \cdot \frac{x_i - {}^2m_{k,i}}{\left(\sigma_{k,i}\right)^2} \tag{A32}$$

$$\frac{\partial f_r^k}{\partial \sigma_{k,i}} = \frac{\overline{w}_r^k \cdot \left[ \left( \overline{f}^k - {}^2\mu_{k,i} \cdot \prod_{l=1, l \neq i}^n \left( \overline{\mu}_{k,l} \right) \right) \cdot \frac{\left(x_i - {}^1m_{k,i}\right)^2}{\left(\sigma_{k,i}\right)^3} \right]}{\overline{w}_r^k + \underline{w}_r^k}$$

$$+ \frac{\overline{w}_r^k \cdot \left[ \left( \overline{f}^k - {}^1\mu_{k,i} \cdot \prod_{l=1, l \neq i}^n \left( \overline{\mu}_{k,l} \right) \right) \cdot \frac{\left(x_i - {}^2m_{k,i}\right)^2}{\left(\sigma_{k,i}\right)^3} \right]}{\overline{w}_r^k + \underline{w}_r^k}$$

$$+ \frac{\underline{w}_r^k \cdot \underline{f}^k \cdot \left[ \frac{\left(x_i - {}^1m_{k,i}\right)^2 + \left(x_i - {}^2m_{k,i}\right)^2}{\left(\sigma_{k,i}\right)^3} \right]}{\overline{w}_r^k + \underline{w}_r^k} \tag{A33}$$

*Convergence Analysis of Learning Algorithm*

The Lyapunov function is used to guarantee learning algorithm convergence. The Lyapunov function is defined as

$$V_p(k) = E_p(k) = \frac{1}{2} e_p^2(k) = \frac{1}{2} \left( t_p(k) - \hat{y}_p(k) \right)^2 \tag{A34}$$

Equation (A35) shows the Lyapunov function changes.

$$\Delta V_p(k) = V_p(k+1) - V_p(k) = \frac{1}{2} \left( e_p^2(k+1) - e_p^2(k) \right) \tag{A35}$$

Next, the moment error is calculated from Equation (A36).

$$e_p(k+1) = e_p(k) + \Delta e_p(k) \cong e_p(k) + \left[ \frac{\partial e_p(k)}{\partial W} \right]^T \Delta W \tag{A36}$$

In Equation (A36), $\Delta W$ is parameter changing, where $W = \left[ \sigma_{k,i}, {}^1m_{k,i}, {}^2m_{k,i}, c_{k,i}, s_{k,i} \right]$. In Equation (A37), the general form of gradient-based updating is presented.

$$W(k+1) = W(k) + \Delta W(k) = W(k) + \eta \cdot \left( -\frac{\partial E_p(k)}{\partial W} \right) \tag{A37}$$

where

$$\frac{\partial E_p(k)}{\partial W} = -e_p(k) \cdot \frac{\partial \hat{y}}{\partial W} \tag{A38}$$

Equation (A35) can be rewritten as Equation (A39).

$$\Delta V_p(k) = \frac{1}{2} \left( e_p^2(k+1) - e_p^2(k) \right) \tag{A39}$$

$$= \frac{1}{2} \left[ \left( e_p(k+1) - e_p(k) \right) \right] \cdot \left[ \left( e_p(k+1) + e_p(k) \right) \right]$$

$$= \frac{1}{2} \Delta e_p(k) \left[ 2 \left( e_p(k) \right) + \Delta e_p(k) \right]$$

$$= \frac{1}{2}\Delta e_p(k)\left[2(e_p(k)) + \Delta e_p(k)\right]$$

$$= \left[\frac{\partial e_p(k)}{\partial W}\right]^T \cdot \eta \cdot e_p(k) \cdot \frac{\partial \hat{y}(k)}{\partial W} \cdot \left\{ e_p(k) + \frac{1}{2}\left[\frac{\partial e_p(k)}{\partial W}\right]^T \cdot \eta \cdot e_p(k) \cdot \frac{\partial \hat{y}(k)}{\partial W} \right\}$$

$$= -\left[\frac{\partial \hat{y}(k)}{\partial W}\right]^T \cdot \eta \cdot e_p(k) \cdot \frac{\partial \hat{y}(k)}{\partial W} \cdot \left\{ e_p(k) - \frac{1}{2}\left[\frac{\partial \hat{y}(k)}{\partial W}\right]^T \cdot \eta \cdot e_p(k) \cdot \frac{\partial \hat{y}(k)}{\partial W} \right\}$$

$$= -\eta \cdot (e_p(k))^2 \left|\frac{\partial \hat{y}(k)}{\partial W}\right|^2 \cdot \left[1 - \frac{1}{2}\eta \cdot \left|\frac{\partial \hat{y}(k)}{\partial W}\right|^2\right]$$

In order for $\Delta V_p(k) < 0$, then:

$$0 < \eta < \frac{2}{max\left|\frac{\partial \hat{y}(k)}{\partial W}\right|^2} \tag{A40}$$

If (A40) holds for every parameter $W = \left[\sigma_{k,i}, {}^1m_{k,i}, {}^2m_{k,i}, c_{k,i}, s_{k,i}\right]$, then the algorithm is definitely convergent.

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
