# Peer review of "Modeling Renewable Energy Systems by a Self-Evolving Nonlinear Consequent Part Recurrent Type-2 Fuzzy System for Power Prediction"

_sustainability, doi:10.3390/su13063301_

Round 1

Reviewer 1 Report

The comments are attached in the document.

Author Response

Reviewer 1

The manuscript titled "Modelling of Renewable energy systems by a self-evolving nonlinear consequent part recurrent type-2 fuzzy system for power prediction" presents a power prediction algorithm using modified NCPRT2FS technique with a couple of case studies on solar and wind prediction applications. However, the current version of the manuscript needs significant modifications for acceptance of publication in the Journal. Few recommendations and comments for authors to consider are as follows:

1.Avoid using Acronyms in the title of the manuscript.

Answer: Thanks to the dear reviewer, this was done.

2.The authors claim to propose a novel analytical model for predicting the power of solar and wind applications. There should be a proper explanation on how this proposed technique is different from the article written by the authors," A class of type-2 fuzzy neural networks for nonlinear dynamical system identification".

Answer: Thanks for your attention, it should be noted that some of the authors of the paper specialize in type-2 fuzzy neural systems and its applications, and naturally have published several articles with the same appearance but completely different interior. The innovations of this article that set it apart from other works are as follows: 1- Using nonlinear functions in the “Then” part of fuzzy rules 2- Introducing a new mechanism in structure learning (while structure learning was not used in previous articles) 3- Using adaptive learning rate (in the previous article, the training rate was fixed) 4- Convergence analysis of the type-2 fuzzy neural network learning algorithm presented in this article (while in previous articles there was no guarantee of convergence or stability) 5- Finally, in this article, some optimization techniques (including pruning fuzzy rules, initial adjustment of membership functions, etc.) are performed, which is due to the experience of previous work. This paragraph has been added to the end of introduction section.

3.There is a lack of proper explanation of the paper's novelty, and the background section needs to highlight the state-of-the-art and indicate how it is different from the existing articles.

Answer: Thanks for your suggestion, in the revised version, the innovations of the article and its differences with the existing works are stated in more detail (See the answer of your previous question).

4.The abstract is too generic, and it is highly recommended to highlight the significance of the work and indicate the overview of the abstract results.

Answer: Thanks for your suggestion, in the revised version, the abstract has been rewritten.

5.The background section is fragile, and the authors need to clarify and explain the difference between the current study with the previously published articles.

Answer: Thanks for your attention, the background section has been revised and rewritten.

6.Text in figures must be presented in high quality. Also, all the parameters in each formula must be introduced.

Answer: Some figures that contained text were modified. Also, the parameters of all formulas were defined and explained.

7.EQ 11 and 12 sums up the explanation on EQ 26-45 what is the need for explaining it separately.

Answer: Equations 11 & 12 are for the feedforward phase. But equations 26 to 45 are related to the parameter update phase (backpropagation).

8.EQ 46-58 are in the previous work of the authors. How are they different in this application? If they are the same, why do we need to include the min this paper?

Answer: Equations 46 to 58 are for updating the "if" part parameters of the fuzzy rules (parameters of membership functions and their weights), which can be common to many articles. The difference between type-2 fuzzy neural networks can be in the "then" part of fuzzy rules, learning algorithm (parametric and structural), number of intermediate layers, normal or recurrent neurons, Mamdani or TSK model, etc. But usually the membership functions are the same in all of them.

  1. How is the proposed technique different from the authors' previous works other than using a new dataset and added rules?

Answer: Please see the answer of your second question.

  1. The type 2 fuzzy system rules can be considered a contribution, but most of the article's writing is like the ones you have already published. I would recommend major reworking in sections of the paper highlighting the work's novelty and the justifications of the fuzzy system's selected rule.

Answer: Thank you for your valuable suggestion. The article was fundamentally rewritten and innovations were highlighted.

  1. The conclusion is more like a summary of the work. So, consider including the highlights of results in this section.

Answer: The conclusion was rewritten.

  1. The authors should indicate the level of confidence in the proposed method.

Answer: Thank you for your reminder, since in this article the convergence (stability) of the training algorithm has been proved mathematically, so we can say with certainty and confidence that the solution presented in this article will be reached to answer. But to find the optimal answer, you have to run the program several times and get the best answer. Also, the root mean square error (RMSE) criterion is presented (Table 3), which shows that a very small and acceptable error has been obtained.

13.In addition to this, the paper needs proper proofreading to make writing more cohesive and convincing. Few formatting errors to be corrected:

1.The authors should check the whole text to correct both grammar and typo mistakes.

Answer: Given that the native language of none of the authors is English, there are naturally spelling and grammatical mistakes. However, the whole text of the article was reviewed many times and many mistakes were resolved.

  1. Avoid group citations in the manuscript.

Answer: Thanks for your notification, in some cases; we have no choice except to group citations. But this issue was resolved as much as possible.

  1. The alignment of the tables is inconsistent.

Answer: The alignment of the tables was corrected.

  1. Many equations are repetitive and not in the format of the Journal.

Answer: Given that we have stated all the details and all the formulas of our method, of course some formulas may be very similar but not duplicate. However, in the revised version, while reviewing all the formulas, an attempt has been made to observe the format of the journal.

  1. The authors should check that all references are appropriately written according to the Journal.

Answer: All references were reviewed and converted to journal format.

Reviewer 2 Report

The paper contains some interesting ideas. The paper has to do with a rather interesting topic.

The novelty is not stated clearly. The problem is not described clearly. Some more details could make this problem a little clearer. The conclusions on the findings and logically stated are based.

The manuscript shows a complete, clear, and well-organized presentation.

References need up-date. I suggest considering this work: Andrukhiv, A.; Sokil, M.; Fedushko, S.; Syerov, Y.; Kalambet, Y.; Peracek, T. Methodology for Increasing the Efficiency of Dynamic Process Calculations in Elastic Elements of Complex Engineering Constructions. Electronics 202110, 40. https://doi.org/10.3390/electronics10010040

I congratulate the authors for good work.

Author Response

Reviewer 3

The paper contains some interesting ideas. The paper has to do with a rather interesting topic. The novelty is not stated clearly. The problem is not described clearly. Some more details could make this problem a little clearer. The conclusions on the findings and logically stated are based.

Answer: Thanks for your compliment; the article was rewritten to make it better.

The manuscript shows a complete, clear, and well-organized presentation.

Answer: Thank you very much.

References need up-date. I suggest considering this work: Andrukhiv, A.; Sokil, M.; Fedushko, S.; Syerov, Y.; Kalambet, Y.; Peracek, T. Methodology for Increasing the Efficiency of Dynamic Process Calculations in Elastic Elements of Complex Engineering Constructions. Electronics 202110, 40. https://doi.org/10.3390/electronics10010040

Answer: Thanks for your suggestion, some new references were read and added to the article.

I congratulate the authors for good work.

Answer: Thank you very much.

Reviewer 3 Report

The paper presents a  type-2 fuzzy system for identification and behaviour prognostication of an experimental solar cell set and a wind turbine. Besides, this work brings forward a technique to acquire an optimal number of membership functions and the corresponding rules. For fuzzification in the first two layers, Gaussian type-2 fuzzy membership functions with uncertainty in the mean are exploited. The third layer comprises rule definition, and the fourth one embeds fulfilment of type reduction. An adaptively rated learning back-propagation algorithm is extended to adjust the parameters ensuring convergence as well. The paper needs language correction. Generally, the work is a good scientific paper. However, before accepting, some shortcomings must be eliminated. The list of comments is as follows:
1. The language correction is needed;
2. Abstract should be rewritten. Some sentences seem to be too strong
3. In the introduction is worthy of writing about Fuzzy sets and their generalization before FS type2. Additionally, other approaches which used FS should be shortly presented as alternative approaches of evaluation, e.g., A new approach to identifying a multi-criteria decision model based on stochastic optimization techniques; Fuzzy model identification using monolithic and structured approaches in decision problems with partially incomplete data; and similar
4. Equations (e.g. (2) and 3) instead of * should be dot
5. Figures 1-2, the quality of the figures must be improved. It is hard to read values etc.
6. There are problems with equations we have to times (1), (2), etc. The numeration must be fixed.
7. Page 5. this information should be presented in Table or itemize.
8. Table 3 should be improved
9. Conclusions must be rewritten. Please add the future research directions

Author Response

Reviewer 2

The paper presents a type-2 fuzzy system for identification and behavior prognostication of an experimental solar cell set and a wind turbine. Besides, this work brings forward a technique to acquire an optimal number of membership functions and the corresponding rules. For fuzzification in the first two layers, Gaussian type-2 fuzzy membership functions with uncertainty in the mean are exploited. The third layer comprises rule definition, and the fourth one embeds fulfilment of type reduction. An adaptively rated learning back-propagation algorithm is extended to adjust the parameters ensuring convergence as well. The paper needs language correction. Generally, the work is a good scientific paper. However, before accepting, some shortcomings must be eliminated. The list of comments is as follows:

1.The language correction is needed;

Answer: Given that the native language of none of the authors is English, there are naturally spelling and grammatical mistakes. However, the whole text of the article was reviewed many times and many mistakes were resolved.

2.Abstract should be rewritten. Some sentences seem to be too strong

Answer: Thank you very much for taking the time to review the article. The abstract was reviewed and rewritten.

3.In the introduction is worthy of writing about Fuzzy sets and their generalization before FS type-2. Additionally, other approaches which used FS should be shortly presented as alternative approaches of evaluation, e.g., A new approach to identifying a multi-criteria decision model based on stochastic optimization techniques; Fuzzy model identification using monolithic and structured approaches in decision problems with partially incomplete data; and similar

Answer: Thanks for your valuable suggestion, the introduction was reviewed and rewritten and some references added.

  1. Equations (e.g. (2) and 3) instead of * should be dot

Answer:  Thanks to the dear reviewer, in most related and similar articles, it has been customary to use the * symbol as a multiplication, but we respect your opinion and replaced the symbol* by the symbol dot.

  1. Figures 1-2, the quality of the figures must be improved. It is hard to read values etc.

Answer: All the figures in the article were examined and replaced with higher quality figures.

6.There are problems with equations we have to times (1), (2), etc. The numeration must be fixed.

Answer: Thanks for your careful examination. The problem corrected.

7.Page 5, this information should be presented in Table or itemize.

Answer:  The information presented in a Table form.

8.Table 3 should be improved

Answer: Thanks for your careful examination. The Table 3 updated.

  1. Conclusions must be rewritten. Please add the future research directions

Answer: Thanks to the valuable suggestion of the honorable reviewer, the conclusion was reviewed and rewritten.

Round 2

Reviewer 1 Report

The authors have revised the manuscript indicating the contribution of the paper to a considerable level, however, the paper needs major modifications and the recommendations for further revisions are as follows:

  • Authors claim the need for explaining the parameter update phase equations separately. But, in my opinion, this falls under the general understanding you have over the type 2 network systems and indicating it is a generic equation would be more sufficient than explaining individually. 
  • The if-then case rules are genric in most cases as the authors indicated, in that case, is there a justification on the need for a step-by-step derivation in the paper?  Too much generic information can be avoided in a scientific article in general. therefore it is recommended to clearly indicate the rules using top-level equations only.
  • There is a lot of similarity in the write up even in the revised version of the manuscript (~70% as suggested by Turnitin) which is not an ideal option for reproducing a previous work, so it is recommended to the authors to improve the writing section of the manuscript significantly in the next revision. 
  • The case studies highlighted in the manuscript plays a major role in the novelty of the paper indicating the application of the improved method in the new application domain. It is recommended to improve the section highlighting this part. 
  • Figure 3 has no label in the revised version. Please check this.
  • Figure 4 still looks shattered and needs a revision.
  • Line 683-684 can be elaborated. 
  • In table 4 there is a cell with "-" this can be avoided by just merging the cell.
  • A proper explanation on the RMSE calculation could be added and justification on considering this method to calculate RMSE can be included in the manuscript.
  • The manuscript still needs major modifications before it is reconsidered for submission.

Author Response

Reviewer1

The authors have revised the manuscript indicating the contribution of the paper to a considerable level, however, the paper needs major modifications and the recommendations for further revisions are as follows:

  • Authors claim the need for explaining the parameter update phase equations separately. But, in my opinion, this falls under the general understanding you have over the type 2 network systems and indicating it is a generic equation would be more sufficient than explaining individually. 

Answer: Our goal in presenting all the formulas was for the reader of the paper to know all the details of the work so that anyone could easily implement or use it if they wanted to. However, we respect the opinion of the reviewer and only the main equations are in the text and other details are provided in the appendix.

  • The if-then case rules are generic in most cases as the authors indicated, in that case, is there a justification on the need for a step-by-step derivation in the paper?  Too much generic information can be avoided in a scientific article in general. Therefore it is recommended to clearly indicate the rules using top-level equations only.

Answer: Thanks a lot for your suggestion, this was done.

  • There is a lot of similarity in the write up even in the revised version of the manuscript (~70% as suggested by Turnitin) which is not an ideal option for reproducing a previous work, so it is recommended to the authors to improve the writing section of the manuscript significantly in the next revision. 

Answer: Thank you for your attention, the paper has been reviewed and rewritten several times. Fortunately, the similarity rate dropped to less than 11%. The similarity percentage file was sent with the revised version.

  • The case studies highlighted in the manuscript plays a major role in the novelty of the paper indicating the application of the improved method in the new application domain. It is recommended to improve the section highlighting this part. 

Answer: Thanks to your valuable suggestion, some changes were made to the text of the paper to emphasize the case studies. The reader of this paper, by seeing these sentences, realizes the purpose and orientation of the paper (Beginning and end of the introduction).

  • Figure 3 has no label in the revised version. Please check this.

Answer: Figure 3 has label, probably not in the previous version. The label of Figure 3 is "The structure of a T2F system ".

  • Figure 4 still looks shattered and needs a revision.

Answer: Thanks for your reminder; the quality of Figure 4 has improved.

  • Line 683-684 can be elaborated. 

Answer: This line number is related to one of the references. I did not understand the detailed explanation of this reference.

  • In table 4 there is a cell with "-" this can be avoided by just merging the cell.

Answer: Table 4 was modified.

  • A proper explanation on the RMSE calculation could be added and justification on considering this method to calculate RMSE can be included in the manuscript.

Answer: Thanks to your valuable suggestion, the RMSE calculation formula has been added to the text of the article.

  • The manuscript still needs major modifications before it is reconsidered for submission.

Answer: Thank you for reading the article carefully, all the authors have reviewed the article several times and tried to fix all the flaws and mistakes.

Reviewer 3 Report

The paper can be accepted in the current form

Author Response

Dear Professor,

Thanks a lot for your time and supportive comments.

With all the best,

Authors

Round 3

Reviewer 1 Report

Dear Authors, 

Thank you for the revised version of the manuscript. The changes made to the manuscript are justifiable and the current version of the paper looks good for acceptance. 

Just a few minor editorial changes are something that you should be looking at fixing before getting the manuscript published. 

The figure and tables from page 9 on wards are a bit off the alignment it would be great if you can fix it before resubmitting it again.